# Predictors of Internalized Stigma in Patients with Schizophrenia in Northern Chile: A Longitudinal Study

**DOI:** 10.3390/healthcare10112269

**Published:** 2022-11-12

**Authors:** Alejandra Caqueo-Urízar, Felipe Ponce-Correa, Alfonso Urzúa, Matías Irarrázaval, Guillaume Fond, Laurent Boyer

**Affiliations:** 1Instituto de Alta Investigación, Universidad de Tarapacá, Arica 1001236, Chile; 2Programa Doctorado en Psicología, Escuela de Psicología y Filosofía, Universidad de Tarapacá, Arica 1010069, Chile; 3Escuela de Psicología, Universidad Católica del Norte, Avenida Angamos 0610, Antofagasta 1270709, Chile; 4Millennium Institute for Research in Depression and Personality, MIDAP, Santiago 8380453, Chile; 5EA 3279—Public Health, Chronic Diseases and Quality of Life—Research Unit, Aix-Marseille University, 13005 Marseille, France

**Keywords:** internalized stigma, self-stigma, schizophrenia

## Abstract

The study aim was to longitudinally assess internalized stigma in a sample of patients from Northern Chile with a diagnosis of schizophrenia along with indicators of patient recovery, including quality of life, psychotic symptomatology, social cognition, premorbid adjustment, and years of untreated psychosis. The 10-year follow-up of stigma measures and predictors were assessed at public outpatient mental health centers in the city of Arica, Chile, during the months of March–July 2012. A total of 26 patients successfully completed the evaluation. The results revealed that, with the exception of the self-stigma subdimension, no clinically significant changes were seen in the trajectories of internalized stigma ratings between baseline and 10-year follow-up, underscoring the importance of assessing global components such as quality of life and premorbid adjustment during the process of internalizing stigma.

## 1. Introduction

In the last 20 years, there has been an increase in research on self-stigma among persons with severe mental illness. The findings of various pieces of research have demonstrated that this stigma is an overlooked but crucial issue in the course of illness for patients with severe mental disorders [1,2,3,4,5]. Patients who accept and internalize the prejudices and unfavorable stereotypes associated with having a severe mental illness develop self-stigmatizing attitudes [6,7].

Patients develop self-stigma when they internalize the prejudices and unfavorable preconceptions associated with having a severe mental illness, and this has a negative impact on many aspects of their lives [8,9,10,11], including increased depressive disorders [12,13,14], suicidal ideation [15], impaired self-esteem [16,17], quality of life [18,19,20], social relationships [21,22], empowerment [2], resilience [23], adherence to treatment [24,25,26,27], and psychiatric care [28,29,30].

In schizophrenia, self-stigma is linked to poor social contact and lower perceptions of social support, which has a negative impact on recovery results [22]. This reduces the chances of clinical and subjective recovery [31], and plays an important role in personal recovery, given that it affects hope and self-esteem [11,32,33]. Increased internalized stigma affects social exclusion and self-marginalization, which makes it challenging to obtain and uphold social roles. This has an impact on functionality as well [34].

According to several longitudinal studies, higher self-stigma at baseline and follow-up predicted less recovery at 1 and 2 years [10]. Other findings showed that relationships between internalized stigma dimensions and sociodemographic factors show that the disease duration is related to the perceived discrimination subscale and that self-stigma and depression correlate strongly over time [13]. In the early stages of psychosis, depressive symptoms may also present as a viable and less stigmatizing initial therapeutic target and a significant predictor of stigma [35].

Cross-sectional studies have already explored the effects of self-stigma. However, results from longitudinal studies remain limited, especially in populations from Latin American countries, where a lack of psychiatrists and other mental health professionals makes it difficult to include self-stigma measures in clinical practice. This study’s objective was to evaluate internalized stigma over time in a sample of Northern Chilean patients with schizophrenia, along with indicators of clinical improvement.

## 2. Methods

### 2.1. Participants

The sample consisted of 26 patients with schizophrenia, who were diagnosed according to the criteria of the International Classification of Diseases 11th Revision (ICD-11) [36]. All participants were users of outpatient mental health services in the city of Arica (Chile), who were evaluated during their monthly follow-up appointments. Each patient had reached a stable stage of their schizophrenia therapy.

Non-probabilistic availability-based sampling was used. To ensure that patients could fully engage in the interviews, a set of exclusion criteria were used to select patients who were in a psychotic crisis or having a sensory or cognitive problem that precluded them from being evaluated. Clinical histories of all patients included medical examinations that ruled out any organic cause for the psychotic symptoms.

### 2.2. Measures

Internalized Stigma of Mental Illness Scale (ISMI-29) [37]: A self-rated questionnaire containing 29 items measured five dimensions, comprised of alienation, endorsement of stereotypes, experience of discrimination, social withdrawal, and resistance to stigma. Each dimension was scored on a four-point scale ranging from 1 (strongly disagree) to four (strongly agree). A higher overall score on the ISMI scale indicates more severe internalized stigmatization. The ISMI has been validated for the Chilean context and its internal consistency and retesting are acceptable for the original version [38]. The original version of the ISMI scale was translated into Spanish by Bengochea-Seco, et al. [39]. Cronbach’s alpha coefficients ranged from (α =0.77 to 0.88).

### 2.3. Clinical Covariates

Positive and Negative Syndrome Scale for Schizophrenia (PANSS) [40]. A 30-item scale that evaluates psychotic symptoms in individuals with schizophrenia assessed 5 dimensions of the disease presentation: positive symptoms, negative symptoms, excitement symptoms, depressive symptoms, and cognitive symptoms [41]. The responses were scored in a 7-level Likert scale (Ranging from 1 = “absent” to 7 = “extreme”). The interpretation of the questionnaire was based on the sum of all responses. The total score was interpreted using the cut-off points of Leucht et al. [42], where a PANSS total score ≥ 58 suggests “mildly ill”, ≥75 to “moderately ill”, ≥95 to “markedly ill” and ≥116 to “severely ill”. The PANSS used has been translated and validated in Spain by Peralta & Cuesta [43] and also Fresán et al. [44] for the Latino population.

Premorbid Adjustment Scale (PAS) [45]. The Premorbid Adjustment Scale evaluates the achievement of normal developmental milestones at certain life stages, prior to the initial onset of psychotic symptoms. Functioning is assessed based on four critical age periods: childhood (0–11 years), early adolescence (12–15 years), late adolescence (16–18 years), and adulthood (19 years and older). Furthermore, five major psychosocial domains are evaluated in the aforementioned life stages: sociability and withdrawal, peer relations, school performance, school adjustment, and socio-sexual adjustment. Note that socio-sexual functioning is not included as a psychological domain during the childhood period, just as school performance and school adjustment are not measured during the adulthood period. The scale comprises 26 items with a score range from 0 to 6, where “0” accounts for normal adjustment and “6” accounts for severe impairment in the domains evaluated. The clinician selects the number that best matches the closest descriptive statement. The overall PAS score is then calculated based on the average scores obtained in each of the developmental subscales and the general section. Higher scores represent lower levels of premorbid adjustment.

The scale was adapted for Spanish speakers by Barajas et al. [46] with acceptable levels of internal consistency on the evaluated domains: total PAS scale (α = 0.89), sociability and withdrawal (α = 0.89), peer relationships (α = 0.89), school performance (α = 0.84), school adjustment (α = 0.86) and sexual adjustment (α = 0.76).

Social Cognition Scale for Psychosis (GEOPTE) [47]. The GEOPTE scale corresponds to a 15-item scale designed to measure cognition in schizophrenia. Each item is scored from one to five in ascending order of severity: (1) none, (2) a little, (3) normal, (4) quite a bit, and (5) very much, with higher values indicating a higher level of functional impairment. The first seven items measure neurocognitive functions, while items eight to fifteen measure social cognitive function. Note that each set of factors is assessed separately to compare neurocognitive deficit (GEOPTE 1–7, range 1–35) and social cognitive impairment (GEOPTE 8–15, range 1–40). The scale revealed good reliability (α = 0.84)

Duration of Untreated Psychosis (DUP) [48]: The DUP period was considered from the onset of the first acute psychotic episode until the establishment of the antipsychotic treatment. To improve precision, temporal evolution was estimated based on the information given by the patient, his or her primary caregiver, and the clinical record.

Sociodemographic Covariates: For a sociodemographic characterization of the sample, the following items were evaluated: ethnicity (1 = not belonging; 1 = belonging to an ethnic group), gender (1 = male; 2 = female), educational level (1 = primary; 2 = high school; 3 = superior), occupation (0 = no occupation; 1 = student; 2 = worker); ncome (1 = below minimum; 2 = greater than minimum), offspring (1 = yes; 0 = no), couple (1 = yes; 0 = no).

### 2.4. Ethical Considerations

The study was conducted in accordance with the Declaration of Helsinki. The clinical record of the information obtained was handled anonymously and informed written informed consent was obtained from the patients and their primary caregivers. The study was approved by the Ethics Committee of the Universidad de Tarapacá (18/2009) and the Chilean National Health Service.

### 2.5. Procedure

Over the course of a three-month period (from March to July 2012), the baseline was assessed in three outpatient facilities in the city of Arica, Chile. When patients went to their subsequent monthly medical checkups, they were encouraged to participate. Patients were assessed for 40–60 min by two psychologists who worked on the research team under the primary investigator’s supervision.

The participants’ informed written consent was acquired. The study’s goals and the participation’s voluntariness were described. The same process used for gathering baseline data was applied to the 10-year follow-up review.

In the beginning, 85 people in total were assessed during the baseline phase, but only 26 people successfully completed the 10-year follow-up. Thus, there was a loss of 59 participants during follow-up, of whom 27 persons died, 15 moved cities and could not be contacted, 10 declined to participate, and 7 presented significant cognitive impairment that prevented them from answering the questionnaire autonomously.

### 2.6. Statistical Analyses

To assess changes in stigma scores, the change between stigma categories for persons in the high and low categories between baseline (Time1; T1) and follow-up (Time2; T2) was considered. Additionally, an individual effect-size approach [49] was used to assess clinically meaningful change. A 0.5 or greater change in the effect size of the internalized stigma scores between baselines and follow up was considered meaningful:(Stigma T2 − Stigma T1)/Standard Deviation of Stigma T1.


All measures at T1 were then examined using linear regression analysis with stigma (T2) as the dependent variable. The stigma (T2) met the assumptions of regression analysis. There was no evidence of collinearity among the predictor variables in any of the analyses. values of <0.05 were considered statistically significant. All statistical analyses were performed using IBM SPSS version 25 software.

## 3. Results

The sample’s mean age was 42.3 years (SD = 11.6), with 16 of 25 patients (61.5%) being men, 20 (76.9%) being single, and 15 (57.7%) self-identifying as members of an ethnic group (Table 1). The age of onset of the first acute psychotic episode was 20.5 years (SD = 7.4), and the age of treatment was 22.4 years (SD = 7.08). Antipsychotic medications, psychotherapy, and occupational therapy were given to all patients. The totality of the sample declares an income below the minimum wage. Only one individual (4%) exhibited severe psychotic symptoms, while 23 (88%) had mild symptoms and two (8%) had marked psychotic symptoms.

When internalized stigma scores were compared between baseline and follow-up, the results showed that there was no clinically significant change in the overall internalized stigma score, but there was a shift between baseline and follow-up in the self-stigma subdimension (d = 0.56). Negative symptoms (d = 0.64) and social cognition (d = 0.53), on the other hand, exhibited a change between baseline and follow-up over the critical value of the effect size (d = 0.5) and were, therefore, clinically significant changes according to the analysis of clinical variables. Table 2 includes information on means, standard deviations, and clinically significant change scores.

The results of the linear regression model (Table 3) containing the baseline covariates as predictors of the total internalized stigma score at follow-up (T2) were significant (*F* = 2.65, *p* < 0.05) and predicted 36% of the total variance. The baseline (T1) covariates with the greatest weight were quality of life (*β* = 0.74, *p* < 0.01) and premorbid adjustment (*β* = −0.54, *p* < 0.05).

In relation to the subdimensions of internalized stigma (Table 3), the covariate model was significant for alienation (*F* = 3.08, *p* < 0.05) and self-stigma (*F* = 2.66, *p* < 0.05). For the case of alienation, the covariates with the highest statistical weight were quality of life (*β* = −0.91, *p* < 0.0) and premorbid adjustment (*β* = 0.51, *p* < 0.0). For the self-stigma regression model, the only significant predictor was quality of life (*β* = −1.01, *p* < 0.0).

## 4. Discussion

The study examined the implications of a set of clinical indicators as predictors of internalized stigma with 10-year follow-up measures, using data from 26 people who werediagnosed with schizophrenia. No clinically significant change in the trajectories of internalized stigma ratings between baseline and the 10-year follow-up was observed. Nevertheless, patients were seen to worsen over time relative to baseline in the self-stigma subdimension, with a clinically meaningful change. The analysis of clinical variables revealed a clinically significant change in negative symptoms (d = 0.64) and social cognition (d = 0.53).

Although there is a substantial body of research pointing to the high prevalence of stigma in people diagnosed with schizophrenia [1,2,3,4,5], and despite the importance of reducing stigma [50,51] especially because of its negative effects on the recovery of people diagnosed with schizophrenia [52,53], it is possible that the results of this study on the absence of clinically significant changes in stigma measures over time was related to previous findings on the limitation and lack of effectiveness of interventions to reduce stigma in patients diagnosed with severe mental disorders, such as schizophrenia [31,54]. Despite the fact that it was not possible to find previous studies that analyzed stigma trajectories with a follow-up of 5 years or more, in this study, clinically significant changes were observed in other relevant variables addressed in previous correlational studies, such as negative symptoms [55] and social cognition [56]. Negative symptoms affect the personal desire for self-exclusion and produce a marked deterioration in interpersonal skills and social connectedness [57]. An improvement in social cognition implies an increase in the ability to understand the social world and an improvement in the experience of feeling connected to others in order to react emotionally appropriately to others [58]. Clinically substantial changes in negative symptoms and social cognition highlight how crucial it is to offer interventions that enhance people’s capacity to relate to and accurately interpret their social environment in order to lessen internalized stigma.

Finally, the results that quality of life and premorbid adjustment predict variations in internalized stigma after a 10-year follow-up are concordant with previous cross-sectional studies that found relationships between quality of life and stigma without being able to establish the nature or direction of the relationship [59]. Nevertheless, the findings of this study converge with previous results indicating that a low quality of life increases stigmatization and the internalization of negative stereotypes [3], which would highlight the need for comprehensive psychological interventions to reduce internalized stigma. Although premorbid adjustment is often considered a predictor of subjective psychological measures, such as internalized stigma, this has mainly been studied in cross-sectional studies [23,60]. In this study, internalized stigma at 10-year follow-up was predicted by premorbid poor adjustment. It is possible that a poor premorbid adjustment severely restricts socio-cognitive development, including the capacity to understand one’s own internal states and the social world, which could make it harder for patients to deal with the diagnosis’ negative connotations or the disorder’s limitations. The findings demonstrate that psychosocial functioning before psychosis influences the patient’s subjectivity and representation of the condition, in addition to having implications for the severity of the disorder [60].

This study has some limitations. The absence of information on internalized stigma during the time between the two interviews is one of the weaknesses of the study. Second, although time was considered in the regression analysis, the interval between follow-up assessments varied greatly. Third, the responses were based solely on self-reports and not on a variety of sources. Fourth, the fact that all participants were from a particular geographic region, with a large number of residents in contact with mental health professionals, may influence the degree of generalizability of the results. Despite the limitation, this is the first longitudinal study of internalized stigma in adults with schizophrenia to be conducted in a Latin American country, and some of the strengths include the use of a multiracial sample, a multivariate analysis, and the ability to identify the independent effects of each predictor variable.

## 5. Conclusions

The results show that the trajectories on the internalized stigma measures are relatively stable over time, with small changes that were not clinically significant except for the self-stigma subdimension. However, the measures of quality of life and premorbid adjustment were significant predictors of follow-up measures of internalized stigma and its subdimensions of alienation and self-stigma. These findings highlight the need to evaluate overall factors such as quality of life and social functioning before the development of psychosis during the internalization phase of stigma. Future research should consider follow-up measurements with shorter time intervals to understand the mechanisms involved in improvements in self-stigma and social cognition. Finally, the results highlight the importance of subjective measures such as quality of life in overcoming stigma.

## Figures and Tables

**Table 1 healthcare-10-02269-t001:** Clinical and Sociodemographic characteristics.

Sociodemographic Characteristics		*n* (%)
Gender	Male	16 (61.5%)
	Female	10 (38.5%)
Ethnicity	Yes	15 (57.7%)
	No	11 (42.3%)
Educational Level	Primary	5 (19.2%)
	High School	19 (73.1%)
	Superior	2 (7.7%)
Occupation	No occupation	20 (76.9%)
	Student	1 (3.8%)
	Worker	5 (19.2%)
Income	Below Minimum	26 (100%)
	Greater than Minimum	-
Have Children	Yes	3 (11.5%)
	No	23 (88.5%)
Couple	Yes	6 (23.1%)
	No	20 (76.9%)
**Clinical characteristics**		** *n (%)* **
Diagnosis (T1)	Schizophrenia	18(69%)
	Residual Schizophrenia.	1(4%)
	Paranoid Schizophrenia	6(23%)
	Schizoaffective Disorder	1(4%)
Principal Antipsychotic	Risperidone	12 (46.2%)
	Aripripazole	2 (7.7%)
	Clozapine	5 (19.2%)
	Fluphenazine Decanoate	2 (7.7 %)
	Olanzapine	5 (19.2%)
Coadjuvant medication 1	Antipsychotic	11 (42.3%)
	Mood stabilizer	1 (3.8%)
	Antidepressant	3 (11.5%)
	Without coadjuvant	11 (42.3%)
Coadjuvant medication 2	Antipsychotic	6 (23.1%)
	Mood stabilizer	1 (3.8%)
	Antidepressant	0
	Without coadjuvant	19 (73.1%)
Psychotherapy (last 12 months)	YES	8 (30.8%)
	NO	18 (69.2%)
Occupational Therapy (last 12 months)	YES	5 (19.2%)
	NO	21 (80.8%)

**Table 2 healthcare-10-02269-t002:** Longitudinal comparison of indices between baseline and follow-up.

		Baseline		Follow-Up		Clinical Change
		Mean	d.s	Mean	d.s	
ISMI	Total Score	69.34	13.31	68.00	12.84	0.10
	Self-Stigma	13.50	3.40	15.42	4.24	0.56
	Alienation	14.69	5.50	13.50	4.01	0.21
	Experience of discrimination	12.84	4.06	12.30	3.91	0.12
	Social withdrawal	15.88	5.36	13.61	4.80	0.36
	Resistance to stigma	12.42	2.90	13.61	4.80	0.14
GEOPTE	Social cognition	22.03	7.18	18.60	5.78	0.53
S-QoL	Quality of life	55.07	11.92	58.03	10.37	0.24
PANSS	Negative symptoms	13.23	7.12	18.76	4.90	0.64
	Positive symptoms	8.53	4.76	7.34	2.29	0.21
	Psychomotor agitation	9.57	5.27	7.46	3.37	0.41
	Depressive symptoms	5.80	3.80	5.88	1.63	0.01
	Cognitive symptoms	6.23	3.22	5.96	2.19	0.07
Duration of untreated psychosis		5.76	8.79	-	-	-
Premorbid adjustment		13.07	4.00	-	-	-

**Table 3 healthcare-10-02269-t003:** Linear regression analysis of baseline predictors of internalized stigma at follow-up (T2).

Model Fit	ISMI Total Score	Alienation	Self-Stigma	Social Withdrawal	Experience of Discrimination	Resistance to Stigma
F(df)	2.65_(11)_ *	3.08_(11)_ *	2.66_(11)_ *	1.33_(11)_	1.61_(11)_	2.79_(11)_
Adj. R2	0.36	0.48	0.39	0.12	0.21	0.44
Assumptions						
W	0.96	0.96	0.95	0.98	0.92	0.96
DW	1.80					
Coefficient	(*β*)					
Positive symptoms	−0.11	0.16	0.24	0.02	−0.43	0.41
Negative symptoms	0.16	−0.37	−0.15	−0.35	0.12	0.46
Cognitive symptoms	0.36	−0.33	−0.47	−0.42	0.03	0.17
Excitement	−0.20	0.29	0.33	0.35	0.04	−0.92
Depressive symptoms	−0.16	0.04	−0.28	0.03	0.57	0.09
Quality of life	0.74 **	−0.91 **	−1.01 **	−0.81 *	−0.24	0.53
Premorbid adjustment	−0.54 *	0.51 *	0.50	0.53	0.30	−0.07
Social cognition	−0.16	0.10	0.41	0.17	0.28	−0.44
Duration untreated psychosis	0.01	0.23	−0.09	−0.03	0.01	−0.21

Note: * *p* < 0.05; ** *p* < 0.01; F = Statistical F; *p* = Significance; adj. R2 = Coefficient R squared corrected; W = Shapiro–Wilks Test; DW = Durbin–Watson Test; *β* = Standardized.

## Data Availability

The data presented in this study are available on request from the corresponding author.

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
