# Peer review of "Predictors of Internalized Stigma in Patients with Schizophrenia in Northern Chile: A Longitudinal Study"

_healthcare, 2022, doi:10.3390/healthcare10112269_

Round 1
Reviewer 1 Report
1- Write down the abbreviation of ICD-11 and why you rely of this?
2- You need to mention how the participants approached to the stable stage of their mental illness. For example, which kind of schizophrenia? Any other mental health condition associated with Schizophrenia? It is unlikely someone has purely schizophrenia without other mental health condition or other physical illness. You may need to design a table includes what I mentioning to you. It would be so beneficial please
3- You need to mention in a table which medications the participants they are on to make them at the stable stage (including non-pharmacological interventions). This is beneficial for people who are interesting in family medicine and psychiatry.
4- ISMI score tool was used in your methods. It is not in English nor the participants language. You mentioned that translated into Spanish. How we know the translation is valid and fair and correct 100% ? what is your evidence?
5- only 26 patients in only 3 centres? Is that enough to establish the conclusion? Why not try other more centres in different places, and in different care places such as mental health centres or psychiatry hospitals?
6- You explained in the discussion that the analysis of clinical variables revealed a clinically significant change in negative symptoms and social cognition. Come back to results, I saw that the way of significance results is opposite in results in between negative symptoms and social cognition. I saw that means of baseline 22.03 and follow-up 18.60 in social cognition whereas means of baseline 13.23 and follow-up 18.76 in negative symptoms. How you interpret these results? And why?
7- What is your recommendation to other clinicians who read your paper, such as what kind of research need to do for further understanding? Please you may mention this part in conclusion
Reviewer 2 Report
This is an important paper addressing a topical issue in the field of mental health. However, the implications of findings for practitioners need to be very clear. Also, the reliability coefficients for each scale should be indicated in the methodology. Last, the authors should consider editing the manuscript to address issues of grammatical/typo mistakes.
Reviewer 3 Report
This study provides a longitudinal evidence of predictors of internalised stigma among patients with schizophrenia. This study makes a meaningful contribution to literature and practice. The methods and results are sound and presented very well. However, in my opinion, I would advise that authors consider providing further rationale for this study beyond providing longitudinal evidence. Why is this study necessary and what difference can its findings make in literature and clinical practice? While the manuscript in its present state is publishable, I think making this addition would further improve the quality of the manuscript.
Required minor correction is to pay closer attention to some punctuation and structuring slips. For instance, section 2.6 (statistical analysis) needs to move down to stand as a sub-heading.
